# The Relationship between Serum Adiponectin, Urinary Albumin/Creatinine Ratio and Type 2 Diabetes: A Population-Based Cross-Sectional Study

**DOI:** 10.3390/jcm11237232

**Published:** 2022-12-05

**Authors:** Shoma Ono, Satoru Mizushiri, Yuki Nishiya, Ayumi Tamura, Kiho Hamaura, Ryoma Ito, Akihide Terada, Jutaro Tanabe, Miyuki Yanagimachi, Kyi Mar Wai, Kaori Sawada, Kazushige Ihara, Makoto Daimon

**Affiliations:** 1Department of Endocrinology and Metabolism, Hirosaki University Graduate School of Medicine, Hirosaki 036-8216, Aomori, Japan; 2Department of Social Medicine, Hirosaki University Graduate School of Medicine, Hirosaki 036-8216, Aomori, Japan

**Keywords:** serum adiponectin concentration, clinical application, urinary albumin to creatinine ratio, type 2 diabetes, cardiovascular disease

## Abstract

The relationship between serum adiponectin concentration (S-Adipo) and various diseases, such as type 2 diabetes (T2D) is conflicting. We hypothesized that the extent of kidney damage in patients with T2D may be responsible for this inconsistency and, thus, examined association between S-Adipo and T2D after consideration for the extent of kidney damage present. Of the 1816 participants in the population-based Iwaki study of Japanese people, 1751 participants with a complete dataset were included. Multivariate logistic regression analyses revealed that low S-Adipo was independently associated with T2D (<0.001), as was high urinary albumin to creatinine ratio (uACR) (<0.001). Principal components analysis showed that the relative value of S-Adipo to uACR (adiponectin relative excess) was significantly associated with T2D (odds ratio: 0.49, *p* < 0.001). Receiver operating curve analyses revealed that an index of adiponectin relative excess the ratio of S-Adipo to uACR was superior to S-Adipo *per se* as a marker of T2D (area under the curve: 0.746 vs. 0.579, *p* < 0.001). This finding indicates that the relationship between S-Adipo and T2D should be evaluated according to the extent of kidney damage present and may warrant similar analyses of the relationships between S-Adipo and other medicalconditions, such as cardiovascular disease.

## 1. Introduction

Adiponectin is an adipocytokine secreted by adipose tissue, it consists of a nitrogen terminal collagen domain and a carboxyl terminal globular domain, and, thus, belongs to the soluble collagen superfamily with homology to complement factor C1q. Adiponectin is abundant in the human bloodstream (1–30 mg/mL), and its serum concentrations paradoxically decrease in obesity, especially with visceral fat accumulation [1,2,3,4,5]. Although the biological functions of adiponectin vary substantially, adiponectin is known to have beneficial metabolic functions, such as insulin-sensitizing, anti-atherogenic, anti-inflammatory properties, and so on [1,2,3,4,5]. Therefore, various studies were conducted on the relationship between serum adiponectin concentrations and various diseases, such as type 2 diabetes (T2D), cardiovascular disease (CVD), cancer, polycystic ovary syndrome (PCOS), and cognitive dysfunction, but with conflicting results [6,7,8,9,10,11,12,13,14,15,16,17,18,19,20,21,22,23]. We hypothesized that a factor influencing this conflicting relationship might be differences in kidney function. To test this hypothesis, we chose to examine the relationship between serum adiponectin concentration and T2D as an example here. In fact, low serum adiponectin concentration has been reported to be a risk factor for the development of T2D in many populations, and, therefore, serum adiponectin concentration has been used to evaluate the risk for future development of T2D in non-diabetic populations [6,7,8,9]. Accordingly, individuals with T2D are expected to have lower serum adiponectin concentrations than non-diabetic individuals. However, since the serum adiponectin concentration is not always low in patients with T2D, serum adiponectin concentration is currently not often measured in the usual clinical setting. Therefore, examining the reasons for this conflicting relationship identified between serum adiponectin concentration and T2D may lead to an accurate clinical application of serum adiponectin concentration. We speculate that the conflicting relationship may be due to differences in the prevalence and severity of diabetic complications in the study sample [10,11,12,13]. In particular, chronic kidney disease (CKD) may have a significant effect on this relationship [5,14,24,25,26].

Serum adiponectin concentration has been shown to be high in patients with CKD; and, particularly, in those with end stage renal disease [24,25,26]. Although this high serum adiponectin concentration in patients with CKD is yet to be explained, reduced clearance and/or catabolism are proposed [14,26]. Namely, a high serum adiponectin concentration may be a consequence of kidney dysfunction. Because patients with T2D often have nephropathy, their serum adiponectin concentration may be affected by the severity of this complication. Thus, patients with T2D may have a low serum adiponectin concentration as part of the pathophysiology of their diabetes but manifest a higher concentration as a consequence of nephropathy.

To test this hypothesis, we aimed to characterize the relationship between serum adiponectin concentration and T2D, with reference to the severity of the kidney damage present. Principal components (PC) analysis using serum adiponectin concentration and urinary albumin to creatinine ratio (uACR) showed a positive association between PC2, representing adiponectin relative excess, and T2D. Then, we found that this ratio of serum adiponectin concentration to uACR (Adipo/uACR) represents a marker of T2D that is independent of the severity of the kidney damage present. As described, the results of the present study may also suggest an explanation for the conflicting relationship identified between serum adiponectin concentration and other conditions, such as CVD, cancer, PCOS, cognitive dysfunction and so on [14,15,16,17].

## 2. Methods

### 2.1. Study Sample

Participants were recruited from residents aged over 20 years living in the Iwaki area, Japan, through a public announcement (the Iwaki study). The Iwaki study is aimed to prevent lifestyle-related diseases and prolong lifespan, with no inclusion and exclusion criteria set for the study. This study is repeated annually in the Iwaki area of Hirosaki city, which is in the Aomori Prefecture of Northern Japan [27,28]. Of the 1816 participants in the Iwaki study performed in 2014–2017, 1771 individuals for whom a complete dataset, including serum adiponectin concentrations and uACR was available, were enrolled in the present study.

This present study was approved by the Ethics Committee of the Hirosaki University School of Medicine (No. 2014-014, 2014-377, 2016-028, and 2017-026) and was conducted according to the principles of the Declaration of Helsinki. Written informed consent was obtained from all the participants.

### 2.2. Characteristics Measured

Blood samples were collected in the morning from a peripheral vein of fasted participants. Urine samples were also collected in the morning. The following parameters were measured: height, body weight, body mass index (BMI), percentage body fat (fat (%)), fasting blood glucose (FBG), glycated hemoglobin (HbA1c), systolic and diastolic blood pressure, serum low-density lipoprotein (LDL)-cholesterol, triglyceride (TG), high-density lipoprotein (HDL)-cholesterol, uric acid, urea nitrogen, creatinine, adiponectin concentrations, and urinary albumin and creatinine concentrations.

### 2.3. Evaluation of Characteristics and Definition of Disease Statuses

Fat (%) was measured by bioelectrical impedance using a Tanita MC-190 body composition analyzer (Tanita Corp., Tokyo, Japan). HbA1c (%) is expressed as the National Glycohemoglobin Standardization Program value. Estimated GFR (eGFR) was calculated using the following formulae published by the Japanese Society of Nephrology, eGFR = 194 × Cr^−1.094^ × Age^−0.287^ for men and 194 × Cr^−1.094^ × Age^−0.287^ × 0.739 for women [29]. All laboratory tests were performed in a commercial laboratory (LSI Medience Co., Tokyo, Japan) according to standard protocols. Briefly, serum adiponectin concentration was measured by latex turbidimetric immunoassay [30]. Urinary albumin and creatinine concentrations were measured by turbidimetric immunoassay and enzymatic assay, respectively. Diabetes was defined using the 2010 Japan Diabetes Society criteria: FBG concentrations ≥ 126 mg/dL [31], or in participants in whom FBG concentration was not measured, HbA1c levels ≥ 6.5%. Participants who were being treated for diabetes were considered to have diabetes. None of the participants were known to have type 1 diabetes. Hypertension was defined using a blood pressure of ≥140/90 mmHg or treatment for hypertension. Hyperlipidemia was defined using an LDL-cholesterol of ≥140 mg/dL or a TG of ≥150 mg/dL. Alcohol consumption (current or not) and smoking habits (never, past or current) were categorized using questionnaires.

## 3. Statistical Analysis

Parametric and non-parametric data are presented as mean ± SDs and number (%), respectively. Factors potentially related to diabetes were identified using univariate and multivariate (for independent associations) logistic regression analyses, and relationships between clinical characteristics were evaluated using linear regression analyses. PC analysis analyzes the correlations between item and variables to identify principal components with high correlation. Therefore, PC analysis was used to discriminate between the absolute serum adiponectin concentration and uACR as a whole (PC1), and their relative ratio (PC2), and therefore to compare the utility of the use of the absolute serum adiponectin concentration and uACR as a whole (PC1), and their relative ratio (PC2). The association of each PC with T2D was also evaluated using multiple logistic regression analysis, with adjustment for factors found to be associated with T2D on univariate regression analysis. Since PC2 may possibly be substituted by the correction of serum adiponectin concentration to uACR, receiver operating characteristic (ROC) curve analyses were performed to determine whether the Adipo/uACR ratio is a superior marker for T2D than serum adiponectin concentration per se and to determine the optimal cut-off values. For these statistical analyses, we log-transformed (log10) the serum adiponectin concentration, uACR, and TG concentration data to approximate normal distributions; *p* < 0.05 was accepted as indicating statistical significance. Analyses were performed using software JMP Pro version 16.0.1(SAS Institute Inc., Cary, NC, USA).

## 4. Results

### 4.1. Clinical Characteristics of the Study Participants

The clinical characteristics of the participants are shown in Table 1. The mean age of the men was 52.9 ± 16.3 years (*n* = 684) and that of the women was 55.1 ± 16.4 years (*n* = 1087). The proportions of hypertension (42.7% in men and 37.8% in women) and T2D (12.9% in men and 8.2% in women) were similar to the national proportions for men and women reported by the Japanese government in 2014 (the proportions of hypertension in men and women of ≥20 years of age were 36.3% and 26.8%, respectively, and those of T2D in men and women aged 50–59 years were 10.5% and 6.5%, respectively) [32]. There are no published national proportion data for hyperlipidemia using the same definition as that used in the present study, although the proportions recorded in the present study (45.2% in men and 45.3% in women) were similar to those reported in other areas of Japan [7,33,34,35].

### 4.2. Factors Influencing the Link between Serum Adiponectin Concentration and Diabetes

The factors associated with T2D were evaluated using univariate logistic regression analyses (Table 2). As expected, low serum adiponectin concentration was found to be positively associated with T2D. In addition, gender, age, fat (%), blood pressures, serum lipid concentrations, and indices of kidney function, such as eGFR and uACR, were also found to be positively associated with T2D. Therefore, the independent association of T2D with serum adiponectin concentration was next evaluated using multivariate logistic regression analysis. This revealed that low serum adiponectin concentration was actually a factor independently associated with T2D (*p* < 0.001), as was high uACR (*p* < 0.001) but not eGFR (*p* = 0.680).

### 4.3. Association of T2D with Serum Adiponectin Concentration and uACR

Both low serum adiponectin concentration and high uACR were found to be independently associated with diabetes, and linear regression analysis revealed a significant correlation between these parameters (β = 0.1402, *p* < 0.001). Therefore, the association of T2D with low serum adiponectin concentration may be better evaluated in conjunction with uACR, and we next aimed to compare the use of serum adiponectin concentration and uACR together with that of their ratio using PC analysis. Two PCs (PC1, absolute value; PC2, adiponectin relative excess) were identified and the association of each of these PCs with T2D was assessed (Figure 1). The PC loadings for serum adiponectin concentration and uACR were positive for PC1, but for PC2, the PC loadings were positive for serum adiponectin concentration and negative for uACR. These findings indicate that PC1 represents the absolute value as a whole, whereas PC2 represents an excess of serum adiponectin concentration, relative to the uACR. Logistic regression analysis with adjustment for the previously identified factors associated with T2D showed that PC2 was associated significantly positively with diabetes (odds ratio [OR]: 0.49, 95% confidence interval [CI]: 0.41–0.60, *p* < 0.001) but that PC1 was not (OR: 0.99, 95% CI: 0.84–1.18, *p* = 0.938) (Table 3).

### 4.4. Adipo/uACR Ratio Is a Superior Marker for T2D to Serum Adiponectin Concentration Alone

Because the PC score varies according to the sample population used, and, therefore, it cannot be used in a standard clinical setting, we defined the ratio of the serum adiponectin concentration (mg/mL) to uACR (mg/grCr) (Adipo/uACR) as a substitute for PC2. A high Adipo/uACR ratio was found to be significantly negatively associated with diabetes after adjustment for multiple factors (OR [per 0.1 increase]: 0.95, 95% CI: 0.93–0.97, *p* < 0.001). ROC curve analyses were performed to determine whether the Adipo/uACR ratio represent a superior marker for T2D than serum adiponectin concentration, and, indeed, the Adipo/uACR ratio (area under the ROC curve [AUC]: 0.7460, sensitivity: 0.6102, specificity: 0.7917, *p* < 0.001) was found to be superior to serum adiponectin concentration (AUC: 0.5785, sensitivity: 0.5367, specificity: 0.6073, *p* = 0.021) (*p* < 0.001 for the comparison) (Figure 2). The optimal cut-off values of these two parameters for T2D were calculated to be 0.6382 and 8.8, respectively, and these values were associated with ORs of 3.66 (95% CI: 2.59–5.17, *p* < 0.001) and 2.35 (95%CI: 1.61–3.43, *p* < 0.001), respectively, after adjustment for multiple factors.

## 5. Discussion

We performed a cross-sectional study of individuals from a general Japanese population in which we used PC analysis to compare the association of the absolute serum adiponectin concentration and uACR (PC1) and their relative concentrations (PC2) with T2D. As shown in Table 3, PC2 was significantly negatively associated with T2D but there was no association between PC1 and T2D after adjustment for multiple factors. We have also proposed the use of Adipo/uACR ratio as a substitute for PC2 because PC2 cannot be evaluated in a standard clinical setting. ROC analysis revealed that the Adipo/uACR ratio is a superior marker for T2D than serum adiponectin concentration *per se* in this general Japanese population. These fact findings suggest that the severity of kidney damage has to be considered when evaluating the relationship between serum adiponectin concentration and diabetes in patients.

These findings may also be relevant to the relationship between serum adiponectin concentration and CVD, which is also controversial [14,15,16,17]. Although serum adiponectin concentration has been shown to be low in patients with CVD in the general population, differing relationships have been identified in patients with CKD; for example, a 1 mg/mL increase in adiponectin concentration has been shown to be associated with a 3% lower risk of CVD in one study [18], but a 6% higher risk in another study [19]. Therefore, consideration of the extent of kidney damage or, specifically, the use of the Adipo/uACR ratio, may assist in the understanding of this phenomenon in the future. Furthermore, the relationships between serum adiponectin concentration and other medical conditions reported to be associated with serum adiponectin concentrations, such as cancer, polycystic ovary syndrome, cognitive impairment and so on [20,21,22,23], may also be reappraised in this light.

Furthermore, since adiponectin has also been shown to be protective against diabetic complications, such as retinopathy and nephropathy in various mouse models [36,37], it would be interesting to evaluate the association between serum adiponectin concentration and diabetic complications in diabetic patients. The use of uACR-adjusted serum adiponectin concentrations, rather than serum adiponectin concentrations themselves, may allow for an accurate assessment of such associations.

The relationships found in this study may explain why individuals with T2D have both lower adiponectin production and increased urinary albumin excretion (from endothelial dysfunction), which are independent of each other but lead to a higher AUC for T2D when expressed as a ratio rather than either variable alone. However, as described in the introduction, diabetic patients do not always have lower serum adiponectin concentrations, but rather often have higher serum adiponectin concentrations, and, therefore, such possibility has to be evaluated in the future.

In the standard clinical setting, eGFR and uACR can be readily calculated for the evaluation of kidney damage. Actually, serum adiponectin concentrations were significantly correlated with both eGFR (β = −0.191, *p* < 0.0001) and uACR (β = 0.141, *p* < 0.0001). In the present study, we used uACR, rather than eGFR, as an index of the extent of kidney damage because, as shown in Table 2, logistic regression analyses showed an association of uACR, but not eGFR, with T2D, independently of multiple potential confounding factors. In general, eGFR decreases slightly or may increases (hyperfiltration) in the early stages of diabetic nephropathy, whereas uACR continuously increases with the worsening of kidney damage, even during the early stages, and most participants in the present study were in such early stages one and two of CKD (98.4%). Therefore, uACR seemed to be more representative of the extent of kidney damage, at least in the present study sample. However, when examining the relationships between serum adiponectin concentration and conditions other than T2D or studying individuals at later stages of CKD, the Adipo/eGFR ratio may be superior to the Adipo/uACR ratio. This possibility remains to be evaluated in the future.

A strength of the present study is that PC analysis was used to evaluate the relationship between serum adiponectin concentration and T2D, while also considering uACR, and we have clearly differentiated the relationship of the relative and the absolute concentrations of serum adiponectin. This finding could only have been made using such an analysis. Furthermore, we studied a relatively large population-based sample and adjusted the analyses for factors that might confound the results.

The study has several limitations. Firstly, we studied participants in a health promotion study rather than the general population *per se*; therefore, they may have been more invested in keeping themselves healthy than the general population. However, the prevalence of lifestyle-related diseases, such as hypertension, hyperlipidemia, and diabetes, did not differ from the published prevalence in Japan. Secondly, because the study was cross-sectional rather than being a cohort study, we could not assess whether the low Adipo/uACR ratio is a superior marker of the future risk for developing diabetes to serum adiponectin concentration itself. Thirdly, because some drugs, such as pioglitazone, are known to affect serum adiponectin concentration, the detailed information of drugs taken by the study participants appeared to be better included as possible confounding factors in this study. However, we did not monitor such information in detail and, thus, could not perform the analysis. In addition, since we started monitoring the drugs taken in 2015, we have some of the information (*n* = 618): among them, 66 were diabetic and only two were taking pioglitazone. Fourthly, since most participants in the present study were in stages one and two of CKD (namely, non-CKD stages), the results observed here may reflect only facts in these early stages of CKD or in those without CKD. Finally, we did not assess the status of the participants with respect to other complications, such as CVD, which may have affected the results.

In conclusion, association of serum adiponectin concentration with T2D should be evaluated in the light of the extent of any concomitant kidney damage because the serum adiponectin concentration relative to a marker of kidney damage was found to be a superior marker to serum adiponectin concentration alone. Further analyses of the relationship between serum adiponectin concentration and other medical conditions, such as CVD, cancer, polycystic ovary syndrome, cognitive impairment and so on, relative to the extent of kidney damage should be conducted to better understand the utility of this parameter as a marker of disease.

## Figures and Tables

**Figure 1 jcm-11-07232-f001:**
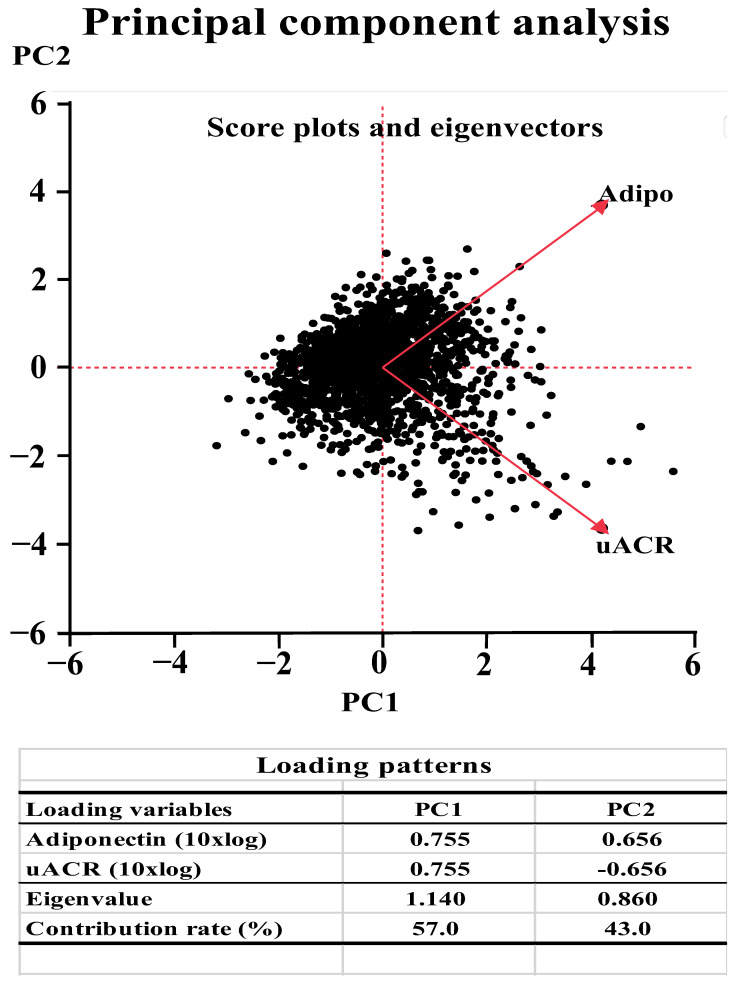
Results of the principal components (PCs) analysis using serum adiponectin concentration (Adipo) and urinary albumin/creatinine ratio (uACR). Eigenvectors with the score plots and loading patterns of PCs are shown in the upper and lower panels, respectively. Adipo and uACR were log-transformed for the analysis to normalize their distributions.

**Figure 2 jcm-11-07232-f002:**
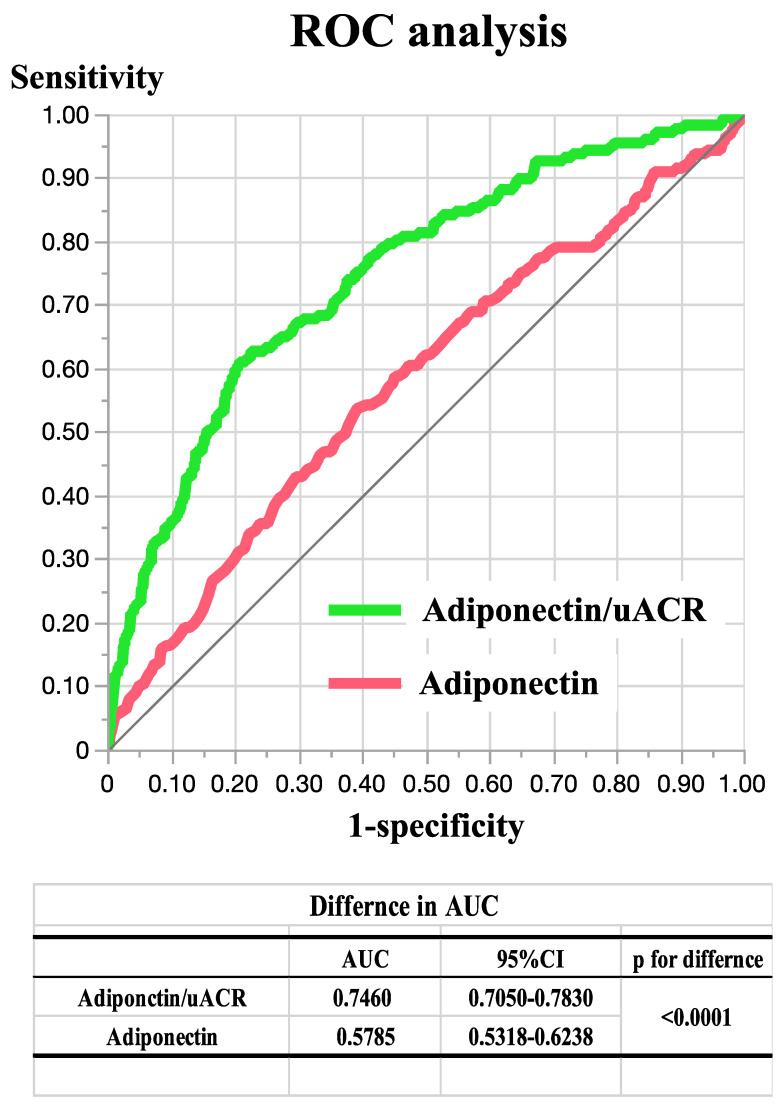
ROC curves of serum adiponectin and serum adiponectin-to-uACR ratio to predict T2D. Red line: serum adiponectin concentration; green line: uACR ratio. The differences in the AUCs between the analyses are shown below the plot. ROC, receiver operating characteristic; T2D, type 2 diabetes; AUC, area under the ROC curve; uACR, urinary albumin/creatinine ratio.

**Table 1 jcm-11-07232-t001:** Clinical characteristics of the participants.

Characteristics	
Number	1771 (684/1087)
Age (yr)	54.3 ± 16.4
Height (cm)	160.4 ± 9.5
Body weight (kg)	67.3 ± 10.7
Body mass index (kg/m^2^)	23.0 ± 3.66
Fat (%)	26.3 ± 8.43
Fasting blood glucose (mg/dL)	91.5 ± 18.0
HbA1c (%)	5.74 ± 0.65
Systolic blood pressure (mmHg)	124.9 ± 19.1
Diastolic blood pressure (mmHg)	73.3 ± 12.1
LDL cholesterol (mg/dL)	115.5 ± 28.5
Triglyceride (mg/dL)	97.9 ± 73.9
HDL cholesterol (mg/dL)	65.0 ± 17.2
Serum uric Acid (mg/dL)	5.03 ± 1.33
Serum urea Nitrogen (mg/dL)	14.58 ± 4.22
Serum creatinin (mg/dL)	0.71 ± 0.27
eGFR (mL/min/1.73 m^2^)	79.9 ± 16.4
Serum Adiponectin (mg/mL)	11.4 ± 6.15
uACR (mg/grCr)	39.2 ± 261.3
Hypertension: *n* (%)	703 (39.7)
Hyperlipidemia: *n* (%)	801 (45.2)
Diabetes: *n* (%)	177 (10.0)
Drinking alcohol: *n* (%)	871 (49.2)
Smoking (Never/Past/Current): *n* (%)	1109 (62.6)/350 (19.8)/312 (17.6)

uACR: urinary albumin to creatinin ratio. HbA1c: glycated hemoglobin. Data are mean ± SD or number of subjects (%).

**Table 2 jcm-11-07232-t002:** Factors correlated with type 2 diabetes.

	Univariate			Multivariate		
Characteristics	OR	95% CI	*p*	OR	95% CI	*p*
Sex (M vs. F)	1.66	1.21–2.26	0.002	1.32	0.91–1.94	0.137
Age (per 1 yr)	1.06	1.044–1.069	<0.001	1.06	1.04–1.07	<0.001
Height (per 1 cm)	0.98	0.96–1.00	0.015	-	-	-
Body weight (per 1 kg)	1.03	1.01–1.04	<0.001	-	-	-
Body mass index (per 1 kg/m^2^)	1.16	1.11–1.20	<0.001	1.11	1.05–1.16	<0.001
Fat (per 1%)	1.04	1.02–1.06	<0.001	-	-	-
Systolic blood pressure (per 10 mmHg)	1.39	1.28–1.50	<0.001	1.04	0.94–1.15	0.475
Diastolic blood pressure (per 10 mmHg)	1.16	1.02–1.32	0.021	-	-	-
LDL cholesterol (per 10 mg/dL)	1.03	0.98–1.09	0.230	-	-	-
Triglyceride (per 1 log mg/dL)	4.81	2.58–8.98	<0.001	1.13	0.49–2.62	0.779
HDL cholesterol (per 1 mg/dL)	0.97	0.96–0.98	<0.001	-	-	-
Serum uric Acid (per 1 mg/dL)	1.12	1.00–1.25	0.054	-	-	-
Serum urea Nitrogen (per 1 mg/dL)	1.11	1.08–1.15	<0.001	1.03	0.99–1.08	0.144
Serum creatinin (per 1 mg/dL)	5.54	2.48–12.38	<0.001	-	-	-
Serum Adiponectin (per 0.1 log mg/mL)	0.88	0.82–0.95	<0.001	0.79	0.72-0.88	<0.001
eGFR (per 10 mL/min/1.73 m^2^)	0.75	0.68–0.83	<0.001	1.03	0.91–1.16	0.680
uACR (per 0.1 log mg/grCr)	1.14	1.11–1.17	<0.001	1.10	1.06–1.13	<0.001
Hypertension	6.01	4.18–8.65	<0.001	-	-	-
Hyperlipidemia	1.83	1.34–2.51	<0.001	-	-	-
Drinking alcohol	0.93	0.63–1.26	0.629	-	-	-
Smoking (Current vs. Never)	0.97	0.63–1.49	0.490	-	-	-

**Table 3 jcm-11-07232-t003:** Association of serum adiponectin concentration with type 2 diabetes, taking into account the uACR.

	Univariate			Multiple Factors Adjusted		
	OR	95% CI	*p*	OR	95% CI	*p*
PC1 (Absolute value) (per 1 PC score)	1.45	1.26–1.66	<0.001	0.99	0.84–1.18	0.938
PC2 (Adiponectin relative excess) (per 1 PC score)	0.42	0.35–0.49	<0.001	0.49	0.41–0.60	<0.001
Adiponectin/uACR (per 0.1 mg/mL/(mg/grCr))	0.92	0.90–0.94	<0.001	0.95	0.93–0.97	<0.001
Adiponectin/uACR (At risk)	5.95	4.30–8.24	<0.001	3.66	2.59–5.17	<0.001
Adiponectin (per 1.0 mg/mL)	0.97	0.94–0.99	0.016	0.95	0.91–0.98	0.001
Adiponectin (At risk)	1.79	1.31–2.45	<0.001	2.35	1.61–3.43	<0.001
uACR (per 10 mg/grCr)	1.01	1.01–1.02	<0.001	1.01	1.00–1.01	<0.001

Factors corrected for: age, sex, and body mass index. uACR, urinary albumin/creatinine ratio.

## Data Availability

All data generated or analyzed during this study are included in this published article.

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
