# Peer review of "The Relationship between Serum Adiponectin, Urinary Albumin/Creatinine Ratio and Type 2 Diabetes: A Population-Based Cross-Sectional Study"

_jcm, 2022, doi:10.3390/jcm11237232_

Round 1

Reviewer 1 Report

The study reports Adiponectin /urinary albumin to creatinine ratio to be superior to serum adiponectin alone for the prediction of T2DM in the Japanese population. The major strength of the study is the large sample size with 1751 participants. The primary component analysis indicate that PC2 representing an excess of serum adiponectin concentration, relative to the uACR is a better predictor of T2DM than PC1 which is the absolute value of Adipo and uACR.

It is evident from the landmark prevention trials that low levels of adiponectin, isolated either from the blood or saliva can predict T2DM in persons at high risk. It is also shown by previous studies that circulating high concentrations of adiponectin to be associated with the pathogenesis of kidney diseases. There are a few suggestions that need to be addressed for better clarity and presentation of the results.

·        -  The inclusion criteria used in the study should be clearly defined and added to the methods. Were there participants without diabetes included in this analysis? 

·      - Ideally, the study population should be selected and grouped as those with CKD and those without for better understanding of the results (98.4% of this study group had CKD).

-In the results section, the term proportion may be used instead of prevalence.

·        - Increased concentration level of adiponectin is not shown.

-It is debatable whether the Adipo/uACR ratio could be used for the normal population without CKD to predict the risk of diabetes. 

Author Response

Reviewer 1

   Thank you very much for having reviewed our manuscript entitled " Clinical application of serum adiponectin concentration is better evaluated after consideration for the urinary albumin/creatinine ratio.: jcm-1994350”. We have changed our manuscript to fulfill your criticisms as much as possible, and in most cases, we just changed as you requested. But, in some cases where another reviewer recommended another way, you may see several changes that you have not suggested or no change that you have suggested. In any case, we beg your generosity to kindly feel satisfied with my responses. The details of the responses to each criticism were written below.

The study reports Adiponectin /urinary albumin to creatinine ratio to be superior to serum adiponectin alone for the prediction of T2DM in the Japanese population. The major strength of the study is the large sample size with 1751 participants. The primary component analysis indicate that PC2 representing an excess of serum adiponectin concentration, relative to the uACR is a better predictor of T2DM than PC1 which is the absolute value of Adipo and uACR. 

It is evident from the landmark prevention trials that low levels of adiponectin, isolated either from the blood or saliva can predict T2DM in persons at high risk. It is also shown by previous studies that circulating high concentrations of adiponectin to be associated with the pathogenesis of kidney diseases. There are a few suggestions that need to be addressed for better clarity and presentation of the results.

------Thanks for your understanding. That is why we conducted this study.

  1. -The inclusion criteria used in the study should be clearly defined and added to the methods. Were there participants without diabetes included in this analysis?

------Surely, I believe your comment is appropriate. There are no inclusion and exclusion criteria for participation, and, thus, most participants were non-diabetic (90.0%) as shown in Table 1. To make this fact more clear we revised the sentences for study sample as follows:

Participants were recruited from the residents aged over 20 years living in the Iwaki area, Japan, through a public announcement (the Iwaki study). The Iwaki study is aimed to prevent lifestyle-related diseases and prolong lifespan with no inclusion and exclusion criteria set for the study.

  1. - Ideally, the study population should be selected and grouped as those with CKD and those without for better understanding of the results (98.4% of this study group had CKD).

--- Yes, your comment is valuable. We had considered such a possibility. However, as we described in the text, most participants in the present study were in stages 1 and 2 of CKD or non-CKD stages (98.4%). Therefore, analysis with participant stratified based on presence of CKD could not be applied in this study sample. Further, we are afraid of some possible miss-understandings in regard with stages of CKD. All individuals are classified into stages 1-5 of CKD according to kidney function, with those in stages 1 and 2 being defined as not having CKD and those in stage3 and above as having CKD. Namely, our study results mainly represent the facts in those without CKD. Since we believe that your suggestion is valuable, we added the following sentences as a limitation.

  Fourthly, since most participants in the present study were in stages 1 and 2 of CKD (namely, non-CKD stages), the results observed here may reflect only facts in these early stages of CKD or in those without CKD.

  1. -In the results section, the term proportion may be used instead of prevalence.

------Upon your suggestion, we changed the term prevalence to the proportion in the results.

  1. -Increased concentration level of adiponectin is not shown.

----- We are not sure what you meant by the comment. Do you mean that increase in serum adiponectin concentrations with progression of kidney damage is not shown? If so, we noticed that we did not show such results, probably because that we thought such results were not directly relevant to the issue reported here. However, we think that your suggestion is valuable, we added the following sentence in the discussion as follows, to shown that serum adiponectin concentration increased significantly with progression of kidney damage.

 Actuary, serum adiponection concentrations were significantly correlated with both eGFR (b= -0.191, p<0.0001) and uACR (b=0.141, p<0.0001).

In another way, if what you meant is the concentration of adiponectin described in the manuscript as excessive. The followings are our reply.

As described in the result, “adiponectin excess” defined here is “an excess of serum adiponectin concentration, relative to the uACR”, and, thus, no concentration of adiponectin per se is considered excessive. However, upon your comment, we realized that such words may mislead the readers, and, thus, we changed the words “adiponectin excess” to “adiponectin relative excess” throughout the text.

In any case, please forgive us, if our explanation is not corresponding to what you meant.

  1. -It is debatable whether the Adipo/uACR ratio could be used for the normal population without CKD to predict the risk of diabetes.

-------This comment appears to be an extension of the comments 2. I think that the above comment is based on the perception that an individual in CKD stages 1 and 2 are affected by CKD. However, as described above, such an individual is not considered as affected by CKD. Individuals in CKD stages 3-5 are defined as affected by CKD, not those in CKD stages 1 and 2. Therefore, our results mainly represent the facts in normal population without CKD. The responses to this comment is the same described for comment 2. So, please understand the situation, and, be generously satisfied with the replies we mentioned here.

Reviewer 2 Report

Comments

-       The material of the article should be more clearly and more fully distributed into the relevant sections. For example, the principal components (PC) analysis, mentioned in the "introduction" section, should be desirable with more details in the "statistical analysis" section.

 -       In the “Methods, Study sample” section, clinical and demographic measurements should go separately from biochemical methods, and biochemical methods should be covered according to an accepted standard with a description of the method, manufacturers and devices.

 -       It is necessary to explain what kind of study was conducted: a cohort cross-sectional observational or prospective with a follow-up period or other?

-       If the study was a cross-sectional study, the design of the study does not allow parameters associated with type II diabetes mellitus to be considered as risk factors. To assess the parameter as a risk factor, a period of observation of outcomes is necessary.

 -       What were the inclusion and exclusion criteria for the study?

 -       It is not clear which concentrations of adiponectin were considered excessive.

 In table 2, the cells must be aligned.

 Clarification of these issues will help to understand the essence of the article

Author Response

Reviewer 2

   Thank you very much for having reviewed our manuscript entitled " Clinical application of serum adiponectin concentration is better evaluated after consideration for the urinary albumin/creatinine ratio.: jcm-1994350”. We have changed our manuscript to fulfill your criticisms as much as possible, and in most cases, we just changed as you requested. But, in some cases where another reviewer recommended another way, you may see several changes that you have not suggested or no change that you have suggested. In any case, we beg your generosity to kindly feel satisfied with my responses. The details of the responses to each criticism were written below.

  1. The material of the article should be more clearly and more fully distributed into the relevant sections. For example, the principal components (PC) analysis, mentioned in the "introduction" section, should be desirable with more details in the "statistical analysis" section.

-----Thanks for your comment. We though that some methodological explanation for PC analysis in introduction may help readers better understand what we did properly. However, we believe that your comment is very appropriate, and, thus, we changed the sentences as follows:

  In introduction.

  Principal components (PC) analysis using serum adiponectin concentration and urinary albumin to creatinine ratio (uACR) showed a positive association between PC2, representing adiponectin relative excess, and T2D. Then, we found that ………

  In statistical analysis.

PC analysis analyzes the correlations between item and variables to identify principal components with high correlation. Therefore, PC analysis was used to discriminate between …….. also evaluated using multiple logistic regression analysis, with adjustment for factors found to be associated with T2D on univariate regression analysis. Since PC2 may possibly be substituted by correction of serum adiponectin concentration to uACR, receiver operating characteristic (ROC) curve ……..

  1. -In the “Methods, Study sample” section, clinical and demographic measurements should go separately from biochemical methods, and biochemical methods should be covered according to an accepted standard with a description of the method, manufacturers and devices.

-----Thanks. Upon your comment, we separated the section describing clinical and demographic measurements from biochemical methods. We described biochemical methods more in details as follows. However, most characteristics measured are so common, we cannot figure out whether detailed description of the method, manufacturers and devices are needed for all these characteristics, provided that this study is not related to any methodological facts. Therefore, we added the methods only for adiponectin and uACR, which are main characteristics of interest in this study, although a description of commercial laboratory measured the all characteristics is described. If you believe that more information for all other characteristics have to be presented, we will do so. Besides, to prepare all such information, we are now asking the commercial laboratory to provide the information, which may take longer than the deadline specified by the editorial office for a response. So, please forgive us for not presenting descriptions of the method, manufacturers and devices for all characteristics measured. In any case, we will respond further:

Briefly, serum adiponectin concentration was measured by latex agglutination nephelometry.Urinary albumin and creatinine concentrations were measured by turbidimetric immunoassay and enzymatic assay, respectively.

  1. -It is necessary to explain what kind of study was conducted: a cohort cross-sectional observational or prospective with a follow-up period or other?

-If the study was a cross-sectional study, the design of the study does not allow parameters associated with type II diabetes mellitus to be considered as risk factors. To assess the parameter as a risk factor, a period of observation of outcomes is necessary.

-----As described, this is a cross-sectional observational study. Although we believe that the words “risk factor” were commonly used in this kind of cross-sectional study, and, thus, were not so odd to be used here. However, we surely think that your comment is appropriate, and, thus, we rephrased the words which may imply cause-effect relationship as much as possible, especially in the sentences which describe results. Namely, we changed the words “risk” and “protection” in result parts throughout the text, although such words were left in some sentences with speculative comments or in discussion. So please forgive us for not rephrasing all such words throughout the text.

4.-What were the inclusion and exclusion criteria for the study?

------Surely, I believe your comment is appropriate. There are no inclusion and exclusion criteria for participation. To make this fact clearer, we revised the sentences for study sample as follows:

Participants were recruited from the residents aged over 20 years living in the Iwaki area, Japan, through a public announcement (the Iwaki study). The Iwaki study is aimed to prevent lifestyle-related diseases and prolong lifespan with no inclusion and exclusion criteria set for the study per se.

  1. -It is not clear which concentrations of adiponectin were considered excessive.

------Thanks for your comment. As described in the result, “adiponectin excess” defined here is “an excess of serum adiponectin concentration, relative to the uACR”, and, thus, no concentration of adiponectin per se is considered excessive. However, upon your comment, we realized that such words may mislead the readers, and, thus, we changed the words “adiponectin excess” to “adiponectin relative excess” throughout the text.

  1. In table 2, the cells must be aligned.

------I am sorry for such miss-alignment, which we did not notice. We corrected them.

Clarification of these issues will help to understand the essence of the article

-----Thanks for your comments

Reviewer 3 Report

The manuscript entitled "Clinical application of serum adiponectin concentration is better evaluated after consideration for the urinary albumin/creatinine

 ratio " is well presented and interested.

limitations:

1. The authors should give more information about adiponectin in the introduction section.

2. The adiponectin is clinically used to be monitored in other diseases,??should be mentioned.

3. The methods needs to be rewritten again to be paraphrased.

4. Discussion should be give some ideas about this topic or the usage of adiponectin in other researches.

Author Response

Reviewer 3

   Thank you very much for having reviewed our manuscript entitled " Clinical application of serum adiponectin concentration is better evaluated after consideration for the urinary albumin/creatinine ratio.: jcm-1994350”. We have changed our manuscript to fulfill your criticisms as much as possible, and in most cases, we just changed as you requested. But, in some cases where another reviewer recommended another way, you may see several changes that you have not suggested or no change that you have suggested. In any case, we beg your generosity to kindly feel satisfied with my responses. The details of the responses to each criticism were written below.

The manuscript entitled "Clinical application of serum adiponectin concentration is better evaluated after consideration for the urinary albumin/creatinine ratio" is well presented and interested.

------Thanks for your evaluation.

limitations:

  1. The authors should give more information about adiponectin in the introduction section.

-----Thanks. We added the following sentences to give more information about adiponectin in the introduction.

    Adiponectin is an adipocytokine secreted by adipose tissue, consists of a nitrogen terminal collagen domain and a carboxyl terminal globular domain, and thus belongs to the soluble collagen superfamily with homology to complement factor C1q. Adiponectin is abundant in huma bloodstream (1-30 ug/mL), and its serum concentrations paradoxically decreases in obesity, especially with visceral fat accumulation. Although the biological functions of adiponectin are varied substantially, adiponectin is known to have beneficial metabolic functions such as insulin-sensitizing, anti-atherogenic, ………

  1. The adiponectin is clinically used to be monitored in other diseases,?? should be mentioned.

-----Thanks. Although we mentioned CVD and diabetic complications as such conditions in the text, upon your suggestion, we added several more conditions reported to be associated with serum adiponectin concentration (although such associations are about to be examined further). Namely, we added the following sentence with some references (added as references, and, the number of reference are changed accordingly) in discussion.

Furthermore, the relationships between serum adiponectin concentration and other disease conditions reported to be associated with serum adiponectin concentrations, such as cancer, polycystic ovary syndrome, cognitive impairment, and so on [32-35], may also be reappraised in this light.

  1. The methods needs to be rewritten again to be paraphrased.

------Thanks. Although we are not sure about how we can paraphrase the methods as you requested, we reorganized the method section with more detailed information added (see the details in the text, as such changes are spread over) .

  1. Discussion should be give some ideas about this topic or the usage of adiponectin in other researches.

---------Thanks for your very important comment. We believe that this comment is related to the comment 2, and, thus, adding other diseases, which may be evaluated in regard with the topic reported here, may be the reply to this comment, as such information surely give some ideas about this topic or the usage of adiponectin in other researches. Therefore, in addition to such comments in discussion, we further added the following words in conclusion.  

    Further analyses of the relationship between serum adiponectin concentration and other disease conditions, such as CVD, cancer, polycystic ovary syndrome, cognitive impairment, and so on, relative to the extent of kidney damage should be conducted

Round 2

Reviewer 1 Report

Generally, serum adiponectin is measured by ELISA or RIA.  A reference is required to show that adiponectin can be estimated by latex
agglutination nephelometry method.

Author Response

Dear Reviewer 1

   Thank you very much for having re-reviewed our manuscript entitled " Clinical application of serum adiponectin concentration is better evaluated after consideration for the urinary albumin/creatinine ratio.: jcm-1994350”. We have changed our manuscript to fulfill your criticism. The details of the responses to the criticism were written below.

Generally, serum adiponectin is measured by ELISA or RIA. A reference is required to show that adiponectin can be estimated by latex agglutination nephelometry method.

-------Thanks. We added the following reference for the methodology, and, then, the reference numbers are changed accordingly throughout the text. Besides, to make the methodology more clear, we changed the name “latex agglutination nephelometry” to “latex turbidimetric immunoassay”.

24. Nishimura A, Sawai Determination of adiponectin in serum using a latex particle-enhanced turbidimetric immunoassay with an automated analyzer. Clin Chim Acta. 2006; 371:163-8.

Reviewer 2 Report

No more comments

Author Response

Dear Reviewer 2

   Thank you very much for having re-reviewed our manuscript entitled " Clinical application of serum adiponectin concentration is better evaluated after consideration for the urinary albumin/creatinine ratio.: jcm-1994350”. We appreciate your following comment for the revised version of the manuscript.

  1. No more comments

      ------------Thanks.